# Contrastive Distant Supervision for Debiased and Denoised Machine Reading Comprehension

**Ning Bian**[1,2], **Hongyu Lin**[2,*], **Xianpei Han**[1,2,*], **Ben He**[1,2], **Le Sun**[1,2]

[1]University of Chinese Academy of Sciences, Beijing, China
[2]Institute of Software, Chinese Academy of Sciences, Beijing, China
bianning21@mails.ucas.ac.cn
{hongyu,xianpei,sunle}@iscas.ac.cn    benhe@ucas.ac.cn

## Abstract

Distant Supervision (DS) is a promising learning approach for MRC by leveraging easily-obtained question-answer pairs. Unfortunately, the heuristically annotated dataset will inevitably lead to mislabeled instances, resulting in answer bias and context noise problems. To learn debiased and denoised MRC models, this paper proposes the Contrastive Distant Supervision algorithm – CDS, which can learn to distinguish confusing and noisy instances via confidence-aware contrastive learning. Specifically, to eliminate answer bias, CDS samples counterfactual negative instances, which ensures that MRC models must take both answer information and question-context interaction into consideration. To denoise distantly annotated contexts, CDS samples confusing negative instances to increase the margin between correct and mislabeled instances. We further propose a confidence-aware contrastive loss to model and leverage the uncertainty of all DS instances during learning. Experimental results show that CDS is effective and can even outperform supervised MRC models without manual annotations.

## 1 Introduction

Machine reading comprehension (MRC) is a fundamental NLP task, which aims to answer questions based on given documents (Hermann et al., 2015; Chen et al., 2016; Seo et al., 2017). Traditional MRC models are usually learned using manually-annotated <question, answer span> pairs (Rajpurkar et al., 2016), which are labor-intensive and time-consuming. To learn MRC models under low-resource settings, distant supervision (DS) is a promising approach that leverages the easily-obtained <question, answer> pairs to heuristically annotate training instances. Specifically, given a <question, answer> pair, a DS-MRC system retrieves relevant passages, matches answers with

---

*Corresponding Authors

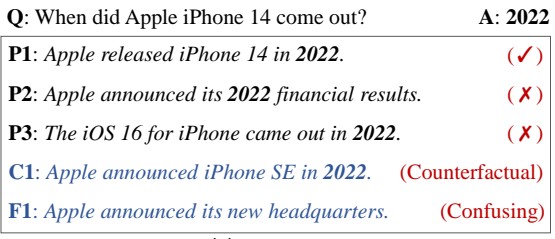

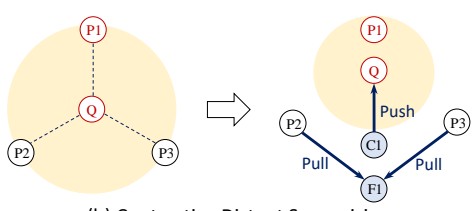

Figure 1: (a) Many DS instances are mislabeled (P2 and P3); (b) During contrastive learning, the counterfactual instance C1 pushes the question towards correct instance P1 by cutting the "When-2022" shortcut, and the confusing instance F1 pulls P2 and P3 away from P1 to due to the similar contexts.

these passages, and uses all matched answer spans to train MRC models (Chen et al., 2017).

Unfortunately, such a heuristically annotation process will inevitably result in numerous mislabeled instances (Min et al., 2019; Lin et al., 2018). As shown in Figure 1(a), by simply matching the answer "2022" with retrieved passages, P2 and P3 are mislabeled instances because they do not express the *release-year* relation of "iPhone 14" in the question. These mislabeled instances make learning effective DS-MRC models a non-trivial task due to the answer bias and context noise problems:

- *Answer Bias problem.* By heuristically annotating all answer spans as correct labels, DS-MRC models tend to make decisions using only answer information, while ignoring the question-context interactions (Ji et al., 2022; Shao et al., 2021). For example, in Figure 1(a), the model may learn to answer "When" questions depending on the When-Date shortcut and ignore all other information such as the interaction between "come

out" in the question and "released" in the context. These spurious correlations cannot generalize well to real-world situations.

- *Context Noise Problem.* Because many instances are noisy, it is difficult for DS-MRC models to identify regular answer patterns for questions. For example, in Figure 1(a), P2 and P3 are mislabeled and the contexts of all three instances are different. This makes it hard for the MRC model to learn the correct "come out"-"released" pattern from such noisy contexts. How to distinguish correct instances from mislabeled instances is critical for distant supervision.

To learn debiased and denoised MRC models, this paper proposes the Contrastive Distant Supervision algorithm – CDS, which can learn to distinguish confusing and noisy instances via confidence-aware contrastive learning. The contrastive learning framework of CDS is shown in Figure 1(b).

To eliminate answer bias, CDS samples counterfactual negative instances and lets DS-MRC models learn to distinguish them from positive instances. Concretely, a counterfactual negative instance is an instance with a correct answer but a wrong counterfactual context. For example, in Figure 1(a), "*Apple announced iPhone SE in 2022.*" is a counterfactual instance because its context does not express the *release-year* relation of "iPhone 14" in the question. By learning to distinguish positive and counterfactual instances, MRC models need to consider both answer information and question-context interaction, and therefore the answer bias can be addressed.

To denoise distantly annotated contexts, CDS samples confusing negative instances, and increases the margin between correct and mislabeled instances by pulling mislabeled positive instances closer to negative instances and pushing correct positive instances away. Concretely, a confusing negative instance contains the wrong answer and a confusing context. For example, in Figure 1(a), F1 is a confusing instance because its context is similar to the mislabeled instances P2 but it does not contain the answer "2022". By learning to distinguish mislabeled instances, CDS can effectively address the context noise problem.

Finally, because there are no golden labels during distant supervision, we propose a confidence-aware contrastive loss, which iteratively estimates the confidence of all instances during training, and optimizes DS-MRC models by taking the uncertainty of training instances into consideration.

We conduct experiments on both close-domain and open-domain MRC benchmarks. Experimental results verify the effectiveness of CDS: On close-domain MRC, CDS significantly outperforms previous DS-MRC methods and achieves competitive performances with supervised MRC methods. On open-domain MRC settings, CDS even outperforms supervised MRC models by a large margin.

The main contributions of this paper are:

- We propose a contrastive distant supervision approach, which can effectively alleviate the answer bias and context noise problems in DS-MRC.
- We design two sampling algorithms and a confidence-aware contrastive loss for DS-MRC models learning: counterfactual instance sampling for answer bias elimination, confusing instance sampling for context denoising, and confidence-aware contrastive loss for uncertainty in distant supervision.
- Experimental results show that CDS is effective and can even outperform supervised MRC models without manual annotations.

## 2 Relate Work

**Distant supervision for MRC**. Distant supervision has been used to train MRC models in low-resource settings, and two main kinds of approaches have been proposed to address the mislabeling problem: (1) filtering noisy labels, and (2) modeling answer spans as latent variables. The noise filtering approaches learn to score and rank DS instances based on answer span positions (Lee et al., 2019; Tay et al., 2018; Swayamdipta et al., 2018; Clark and Gardner, 2018; Lin et al., 2018; Joshi et al., 2017; Chen et al., 2017), question-passage similarities (Hong et al., 2022; Qin et al., 2021; Shao et al., 2021; Deng et al., 2021) and model confidences (Chen et al., 2022; Zhu et al., 2022). The latent variable-based approaches jointly train MRC models and identify correct answer spans using hard-EM algorithms (Zhao et al., 2021; Min et al., 2019; Cheng et al., 2020). In this paper, we address the mislabeling problem via contrastive learning, so that DS-MRC models can be debiased and denoised by sampling counterfactual and confusing negative instances and designing confidence-aware contrastive loss.

**Contrastive learning** is a promising technique that learns informative representations by contrasting positive sample pairs against negative pairs

(Chen et al., 2020). Some studies have leveraged contrastive learning on weak labels: Hong and Yang (2021) contrastively debiases noisy labels by regularizing feature space. Wang et al. (2022a) learns better-aligned representations of partial labels via contrastive learning. Chen et al. (2022) proposes an answer-aware contrastive learning approach to address the translation noise in cross-language MRC. Wang et al. (2022b) propose to exploit span-level contrastive learning for few-shot MRC. Zhang et al. (2021) proposes to sample counterfactual instances and learn inter-modality relationships by contrastive learning for visual commonsense reasoning. Contrastive learning is also widely used for weakly supervised learning (Du et al., 2022; Yan et al., 2022; Gao et al., 2022; Xie et al., 2022; Li et al., 2022b; Zheng et al., 2022; Wan et al., 2022; Li et al., 2022a; Yi et al., 2022; Yang et al., 2022; Li et al., 2021; Chen et al., 2021). In this paper, we leverage contrastive learning to address the bias and the noise problems in DS-MRC.

## 3 Background

**Base MRC Model.** The MRC model used in this paper is

$$\mathcal{P}_{start}, \mathcal{P}_{end} = M(q, p) \qquad (1)$$

where $q$ and $p$ are the question and the passage, respectively, and the model outputs two probability distributions $\mathcal{P}_{start}$ and $\mathcal{P}_{end}$ representing the probabilities of each token being the start and the end positions of the answer span. Our confidence-aware contrastive learning algorithm can be used in any MRC model, and this paper uses the BERT-based MRC model, the same as Devlin et al. (2019).

**Vanilla Distant Supervision for MRC.** Given a <question, answer> pair, the DS-MRC method first retrieves $K$ relevant passages from a corpus. Following Karpukhin et al. (2020), this paper uses the Wikipedia corpus and splits all texts into 100-token chunks. Then, we heuristically match the answer with retrieved passages, and all matching answer spans are used to form <question, passage, answer span> instances. The passages that do not contain the answer are labeled as negative passages. Finally, all DS instances are used to train MRC models as in Chen et al. (2017).

As shown above, the main drawback of vanilla distant supervision for MRC is that many DS instances are mislabeled, resulting in answer bias and context noise problems. In the next section,

we describe how to resolve these problems via contrastive distant supervision.

## 4 Contrastive Distant Supervision for Machine Reading Comprehension

In this section, we present our contrastive distant supervision (CDS) algorithm, which can learn to distinguish confusing and noisy instances via confidence-aware contrastive learning. Specifically, CDS samples counterfactual negative instances to eliminate answer bias, and samples confusing negative instances to denoise distantly annotated contexts. Then, a confidence-aware contrastive loss is proposed to estimate and leverage the uncertainty of all instances during learning. In the following, we describe them in detail.

### 4.1 Counterfactual Negative Instance Sampling

By simply annotating all matched answer spans as correct labels, DS-MRC models tend to make decisions using only answer information, while ignoring the question-context interactions. To eliminate this answer bias, CDS samples counterfactual negative instances and ensures that DS-MRC models can distinguish them from positive instances. Concretely, a counterfactual instance is an instance with the correct answer but a counterfactual context irrelevant to the question. Figure 1(a) shows a counterfactual instance C1 which does not express the correct *release-year* relation of "iPhone 14" but with matched answer span "2022".

Specifically, we sample counterfactual instances from low-confidence DS instances via a rank-based sampling algorithm. The basic assumption is that if the MRC model predicts lower answer probability for an instance, the instance is less relevant to the question (Qi et al., 2021; Izacard and Grave, 2021). Based on this assumption, for a <question, answer> pair, we first predict the answer probability $\mathcal{P}(x)$ of each DS training instance $x$ using the current MRC model, where $\mathcal{P}(x)$ is the $n_{best}$-softmax probability of the answer span as the same as Cao et al. (2020). Then, for each training instance $x$, we sample instances that have significantly lower probabilities than $x$ as its counterfactual instances $\mathcal{U}_x^{CF}$:

$$\mathcal{U}_x^{CF} = \{s_j | \mathcal{P}(s_j) < \mathcal{P}(x) - m\} \qquad (2)$$

where $s_j$ represents a counterfactual instance, and $m$ is the probability margin. We set $m =$

$\max_x(\mathcal{P}(x)) - \text{median}_x(\mathcal{P}(x))$ for flexibility on different $\mathcal{P}(x)$ distributions.

After sampling $\mathcal{U}_x^{CF}$, the answer bias can be addressed by learning to distinguish counterfactual instances contrastively. Firstly, given a training instance $x$ and an MRC model $M$, we obtain the current representations of the question, answer, and counterfactual instance correspondingly as $\boldsymbol{x}^q$, $\boldsymbol{x}^a$, and $\boldsymbol{s}_j^a$. Then, we minimize the distance between the question $\boldsymbol{x}^q$ and the answer $\boldsymbol{x}^a$, while maximizing the distances between question $\boldsymbol{x}^q$ and the counterfactual negative instances $\boldsymbol{s}_j^a$ via the debias loss in Section 4.3.

Because all counterfactual instances are matched with the answer span but are negative, the counterfactual instances will ensure DS-MRC models to answer questions based on both answer information and question-context interaction. In this way, the answer bias can be resolved. For example, as shown in Figure 1(b), the counterfactual negative instances will cut the "When-Date" shortcut, and push the question representation towards the correct instance P1, meanwhile away from mislabeled P2 and P3.

## 4.2 Confusing Negative Instance Sampling

Because many DS instances are noisy, it is difficult to learn to identify regular answer patterns for questions. To denoise contexts, CDS samples confusing negative instances which are used to pull mislabeled positive instances closer to the confusing negative samples. In this way, the margin between correct and mislabeled instances will increase. Concretely, a confusing negative instance should not contain the correct answer but should have a similar context to mislabeled instances. For example, Figure 1(a) shows a sampled confusing instance F1, which is similar to the mislabeled instances P2 and P3 but does not contain the answer.

To sample confusing negative instances, we design a retrieval-based algorithm. Specifically, for a training instance $x$, we retrieve its similar passages, filter out passages with answers, and use the top-$K$ remaining passages as the confusing negative instances $\mathcal{U}_x^C$:

$$\mathcal{U}_x^C = \text{Top}_{p \in \{p_j | a \notin p_j\}}^K \text{sim}(x, p_j) \quad (3)$$

where $p_j$ represents a confusing instance, $a$ is the answer, and $\text{sim}(x, p_j)$ is the similarity function in passage retrieval (this paper uses BM25).

After sampling $\mathcal{U}_x^C$, we learn to denoise contexts by learning to distinguish confusing instances. Given a mislabeled instance $x$, we minimize the distances between its context representation $\boldsymbol{x}^c$ and the representations of confusing negative instances $\boldsymbol{p}_j$, while maximizing the distances between context $\boldsymbol{x}^c$ and the context representations of positive instances $\boldsymbol{x}_j^c$ via a denoise loss in Section 4.3.

By sampling confusing negatives and pulling mislabeled instances closer to confusing negatives, the context noise problem can be effectively addressed. For example, as shown in Figure 1(b), the confusing negative instances will pull the mislabeled instances P2 and P3 closer to confusing negative F1, therefore the influence of these noisy instances will be reduced.

## 4.3 Confidence-Aware Contrastive Learning

Traditional contrastive learning usually needs golden labels of positive and negative instances. Unfortunately, this is not the case in distant supervision, where all labels are uncertain. To address this label uncertainty problem, our contrastive distant supervision algorithm designs a confidence-aware contrastive loss, which iteratively estimates the confidence of all instances during training and optimizes DS-MRC models by taking the uncertainty of instances into consideration. In the following, we introduce how to optimize DS-MRC models using the confidence-aware contrastive loss and how to estimate the confidence scores during training.

**Confidence-aware contrastive loss.** To take into consideration the label uncertainty in DS-MRC, the confidence-aware contrastive loss uses confidence to weight the importance of all instances. The overall loss is:

$$L_{CDS} = \sum_x (\alpha L_{debias} + \beta L_{denoise} + L_{CE}) \quad (4)$$

where $L_{debias}$ is the debias loss used to ensure the model can distinguish counterfactual negative instances, $L_{denoise}$ is the denoise loss used to denoise contexts by distinguishing confusing negative instances, $L_{CE}$ is cross-entropy loss used to learn the answer span prediction ability, $\alpha$ and $\beta$ are hyper-parameters. Given a training instance $x$, we describe these losses in the following.

**Debias Loss**. The debias loss of $x$ is used to ensure the MRC models can distinguish $x$ from its corresponding counterfactual instances:

$$L_{debias}(x) =$$

$$- w_x log \frac{\exp(\boldsymbol{x}^q \cdot \boldsymbol{x}^a)}{\exp(\boldsymbol{x}^q \cdot \boldsymbol{x}^a) + \sum_{s_j \in \mathcal{U}_x^{CF}} \exp(\boldsymbol{x}^q \cdot \boldsymbol{s}_j^a)}$$

$$(5)$$

where $w_x \in [0, 1]$ is the confidence score of the instance $x$. In this way, the loss will be dominated by highly confident correct instances, and the influence of uncertain DS instances will be reduced. By minimizing this loss, the question representation $\boldsymbol{x}^q$ is pushed towards the representations of high-confidence instances $\boldsymbol{x}^a$ and away from the representations of counterfactual instances $\boldsymbol{s}_j^a$.

**Denoise Loss.** The denoise loss of $x$ is used to ensure the MRC models can identify mislabeled instances via the relations to its confusing negative instances:

$$L_{denoise}(x) = -(1 - w_x)$$

$$log \frac{\sum_{p_j \in \mathcal{U}_x^C} \exp(\boldsymbol{p}_j \cdot \boldsymbol{x}^c)}{\sum_{p_j \in \mathcal{U}_x^C} \exp(\boldsymbol{p}_j \cdot \boldsymbol{x}^c) + \sum_{j=1}^J w_{x_j} \exp(\boldsymbol{x}_j^c \cdot \boldsymbol{x}^c)}$$

$$(6)$$

where $J$ is the total number of instances of the <question, answer> pair. This loss is dominated by low-confidence mislabeled instances using the $1 - w_x$ term, and therefore it will not influence the high-confident instances. By minimizing this loss, the context representation $\boldsymbol{x}^c$ of low-confidence instances is pulled towards the confusing instance representations $\boldsymbol{p}_j$ and pushed away from the high-confidence instances $\boldsymbol{x}_j^c$ selected by the $w_{x_j}$ term at the bottom.

**Cross-Entropy Loss.** To learn the answer span prediction ability for MRC, we design a confidence-aware cross-entropy loss:

$$L_{CE}(x) = -w_x( \log \mathcal{P}_{start}(l_x^{start}) + \\ \log \mathcal{P}_{end}(l_x^{end})) \quad (7)$$

where $l_x^{start}$ and $l_x^{end}$ are the start and end positions of the answer span of instance $x$. This loss function is weighted by the confidence $w_x$ for each instance, so that high-confidence instances will have larger effects on the learning of answer span prediction.

**Confidence Estimation.** To model the uncertainty of DS instances, we estimate their confidence by deriving confidence evidence from the current MRC model. Specifically, we employ an iterative estimation strategy: we start with initial confidence scores and iteratively update them during training.

Given an instance $x$ for a question $q$, we use its question-passage matching score to initialize its confidence score. The intuition here is that the matching scores measure the relevance between the question and the passages, providing a reasonable start for confidence. Specifically, we use the BM25 score during retrieval and normalize the confidence scores of all instances of the same question $q$:

$$w_{x_i}^0 = \frac{\text{sim}(q, x_i)}{\sum_{j=1}^J \text{sim}(q, x_j)} \quad (8)$$

where $w_x^0$ is the initial confidence score for the $i$-th instance of the <question, answer> pair, and $\text{sim}(q, x_j)$ is the BM25 matching function.

To update the confidence during training at each time step $t$, we apply a moving-average strategy and update $w^t$ as:

$$w_{x_i}^t = \lambda w_{x_i}^{t-1} + (1 - \lambda)z_i^t \quad (9)$$

where $z_i^t$ is the confidence evidence derived from the current MRC model, and $\lambda \in (0, 1)$ is a constant for moving average. The moving-average strategy smoothly updates the estimated confidence and ensures training stability.

To get model confidence score $z^t$, we design three approaches including *Soft Weighting*, *Hard Max*, and *Positive Average*.

For *Soft Weighting* (SW), $z_i$ is computed as:

$$\mathcal{P}_i = \mathcal{P}_{start}(l_i^{start}) \times \mathcal{P}_{end}(l_i^{end}) \quad (10)$$

$$z_i = \frac{\mathcal{P}_i}{\sum_{j=1}^J \mathcal{P}_j} \quad (11)$$

where $\mathcal{P}_i$ is the predicted span probability for the $i$-th instance. This approach assigns a soft weight for each instance according to its predicted probability by the MRC model.

For *Hard Max* (HM), $z_i$ is computed as:

$$z_i = \begin{cases} 1 & if \ i = \arg \max_{1 \le k \le J} \mathcal{P}_k \\ 0 & else \end{cases} \quad (12)$$

where $\mathcal{P}_k$ is calculated through Equation 10. This approach assumes that only one of the instances for a <question, answer> pair is correct, and picks the one that has the highest probability as the correct instance.

For *Positive Average* (PA), $z_i$ is computed as:

$$\hat{z}_i = \begin{cases} 1 & if \arg\max_c \mathcal{P}_{start}(c) = l_i^{start} \\ & and \arg\max_c \mathcal{P}_{end}(c) = l_i^{end} \\ 0 & else \end{cases} \quad (13)$$

$$z_i = \frac{\hat{z}_i}{\sum_{j=1}^{J} \hat{z}_j} \quad (14)$$

This approach is inspired by Zhao et al. (2021) that if the span predicted by the model matches the labeled answer span, the instance is marked as correct.

Through iteratively refining the confidence of all instances and optimizing the confidence-aware contrastive loss, CDS can estimate and exploit the label uncertainty of DS instances. Therefore the uncertainty problem can be effectively addressed.

## 5 Experiments

### 5.1 Experiment setup

**Datasets.** We evaluate our method on two standard MRC datasets: NaturalQuestions and TriviaQA. NaturalQuestions (NQ) (Kwiatkowski et al., 2019) is an MRC benchmark with manually annotated passages and answer spans. In this paper, we only use the answer strings for training. We use the modified NQ dataset provided by Sen and Saffari (2020), which is a subset of NQ and the MRC task is defined as finding the short answer in the long answer. This offers a more standard context for MRC tasks, whereas the official NQ dataset involves reading long documents or multi-document retrieval, which does not align with our focus. Therefore, the modified NQ dataset provides a suitable setting for evaluating our model's performance. TriviaQA (Joshi et al., 2017) is a DS-MRC dataset collected from trivia quiz websites without golden passages and answer spans[1].

**Baselines.** We compare our method with state-of-the-art supervised and DS-MRC methods. For supervised MRC, we use the original training datasets to train MRC models. For DS-MRC, we compare with two types of methods: noise filtering methods (Xie et al., 2020; Swayamdipta et al., 2018) and a latent variable-based method (Min et al., 2019). We further compare with DS-Top (Chen et al., 2017), which matches overlaps between questions and contexts and picks top-scored answer spans for

---

[1]The licenses of these datasets are Apache License 2.0.

training, DS-All, which uses all individual DS instances to train the MRC model, and DS-Average, which assigns an averaged confidence vector to all instances for a <question, answer> pair. For open-domain DS-MRC, we compare with DistDR (Zhao et al., 2021) which iteratively improves a dense passage retriever through an EM approach, and two state-of-the-art open-domain MRC methods, DPR (Karpukhin et al., 2020) and ANCE (Xiong et al., 2021).

**Training configurations.** We initialize our model using the pre-trained BERT-large model (Devlin et al., 2019). Our model is trained with a learning rate of 2e-5, and the batch size is 32. The model is trained for 3 iterations. We pick the top 20 retrieved passages for training and evaluation. We take $K = 5$ for confusing instance sampling. We set $\alpha = 1$ and $\beta = 1$ for the contrastive loss. For confidence estimation, the moving-average factor $\lambda$ is 0.8.

### 5.2 Close-Domain MRC Results

We first evaluate our CDS method under the close-domain MRC setting, where the passages are provided in the dataset. We use the exact match (EM) and F1 scores of the output answers as in Rajpurkar et al. (2016) for evaluation. From the results shown in Table 1, we find that:

**Confidence-aware contrastive loss is effective for distant supervision.** In experiments, all three confidence estimating approaches CDS-SW, CDS-HM, and CDS-PA achieve good performances in Table 1, and CDS-PA is slightly better than the other two approaches. These results verify that it is important to take label uncertainty into consideration for effective distant supervision.

### 5.3 Open-Domain MRC Results

This section evaluates our CDS method under the open-domain MRC setting. In the first experiment, We use retrieval results (BM25 and DPR-Positive) from open-domain MRC as passages, so that the robustness of different MRC models under noisy retrieved passages can be assessed. Specifically, for BM25, we use the top-20 retrieved results as the passages. For DPR-Positive, we use the "positive contexts" provided by Karpukhin et al. (2020) as the passages. In the second experiment, we apply an ensemble of our three MRC models (CDS-SW, -HM, -PA) in an open-domain MRC pipeline. We use DistDR (Zhao et al., 2021) as the retrieval sys-

| Model | NQ | | | TriviaQA | | |
|---|---|---|---|---|---|---|
| | Close | BM25 | DPR-Positive | Close | BM25 | DPR-Positive |
| Supervised Model | 63.01/75.50 | 11.91/21.50 | 61.40/74.28 | 71.50/76.56 | 43.97/48.36 | 29.16/30.87 |
| Swayamdipta et al. (2018) | –/– | –/– | –/– | 51.60/56.00 | –/– | –/– |
| Hard-EM (Min et al., 2019) | –/– | –/– | –/– | 66.90/ – | 50.70/ – | –/– |
| DS-Top (Chen et al., 2017) | 50.37/64.44 | 14.92/23.29 | 63.10/72.72 | 70.64/74.38 | 48.86/51.15 | 58.54/61.50 |
| DS-All | 48.71/63.30 | 14.21/23.26 | 69.14/77.99 | 78.37/80.69 | 49.94/52.97 | 63.36/65.25 |
| DS-Average | 49.54/63.79 | 13.61/23.10 | 69.26/78.15 | 77.39/79.99 | 49.99/53.00 | 62.97/65.16 |
| Xie et al. (2020) | –/– | –/– | –/– | 54.40/60.20 | –/– | –/– |
| CDS-SW | 50.73/64.97 | 15.49/24.95 | 70.24/78.91 | **78.57/81.29** | 55.20/58.57 | 63.80/66.04 |
| CDS-HM | 51.93/65.60 | 15.85/25.24 | **71.16/79.56** | 77.76/80.50 | 57.15/60.24 | 64.23/66.25 |
| CDS-PA | **52.16/65.71** | **16.40/25.49** | 70.65/78.81 | 78.52/81.18 | **57.96/61.03** | **64.86/66.85** |

Table 1: Evaluation results (EM/F1) on the development sets of NQ and TriviaQA under close-domain MRC setting (Close) and open-domain MRC setting with passages from BM25 and DPR-Positive (Karpukhin et al., 2020) retrieval results.

| Model | Exact Match |
|---|---|
| Supervised DPR | 41.5 |
| Supervised ANCE | 46.0 |
| Distantly Supervised DPR | 37.9 |
| Distantly Supervised ANCE | 38.3 |
| DistDR (Zhao et al., 2021) | 40.5 |
| DistDR + CDS | 40.9 |

Table 2: Evaluation results for open-domain DS-MRC pipeline on the test set of NQ.

tem because it is trained under the distant supervision setting, which is consistent with ours. Table 1 and Table 2 show the results, and we can see that:

(1) **By designing debiasing and denoising strategies, our MRC models can achieve robust performances on noisy passages, and even outperform supervised models.** As shown in Table 1, our models significantly outperform the supervised models in both BM25 and DPR-positive settings. The highest improvement is on TriviaQA with DPR-Positive: 35.70 points on EM and 35.98 points on F1 score. On TriviaQA, our CDS models achieve significant improvements compared with both noise filtering and latent variable-based DS-MRC methods. These results show that noise is a critical problem for DS-MRC models, and CDS can effectively learn debiased and denoised DS-MRC models, therefore improving the robustness of MRC models under the noisy open-domain MRC setting.

(2) **Contrastive distant supervision can be an on-the-fly plug-in for improving the open-domain DS-MRC system.** As shown in Table 2, using our model as the reader module in the DistDR (Zhao et al., 2021) pipeline (DistDR + CDS) achieves higher performance than existing distantly supervised open-domain MRC systems and further minimizes the gap between distant and full supervision. This shows that contrastive distant

supervision is model-free, which can be directly used in existing open-domain MRC systems and complements other promising techniques.

### 5.4 Ablation Studies

We conduct ablation experiments in Table 3 to further show the effectiveness of our method designs.

**Effect of counterfactual negative instance sampling.** We study the effect of counterfactual negative instance sampling by removing the debias loss $L_{debias}$ from the overall contrastive loss. As shown in Table 3, there are performance drops in all three settings compared with the full method, especially in the BM25 setting. These results show that the answer bias problem is critical for DS-MRC models, and our counterfactual instance sampling is effective for addressing the answer bias.

**Effect of confusing negative instance sampling.** We study the effect of confusing negative instance sampling by removing the denoise loss $L_{denoise}$ from the overall contrastive loss. As shown in Table 3, this results in significant performance drops compared with our full method. This verifies that context noise is a critical problem in DS-MRC, and sampling and contrasting with confusing negative instances is effective for denoising distantly annotated contexts.

**Effect of confidence-aware contrastive loss.** We replace our confidence-aware contrastive loss with traditional contrastive loss by simply using instances that contain answers as positive samples and using passages without answers as negative samples. This results in severe performance drops, especially in the close-domain and BM25 settings. We believe this is because too many positive samples are actually mislabeled in this case, which

| Model | Close | BM25 | DPR-Positive |
|---|---|---|---|
| CDS-PA | 78.52/81.18 | 57.96/61.03 | 64.86/66.85 |
| w/o debias contrastive loss | 78.24/80.90 | 53.72/56.96 | 63.43/65.52 |
| w/o denoise contrastive loss | 77.75/80.19 | 56.31/59.41 | 63.66/65.71 |
| w/o confidence-aware contrastive learning | 74.14/77.92 | 52.05/54.29 | 63.78/66.06 |

Table 3: Ablation results (EM/F1) on the TriviaQA dataset.

---

**Question**: What is the capital of Nicaragua in Spanish?

**Answer**: **Managua**

**Correct instance**: Nicaragua is the largest country in the Central American isthmus...**Managua** is the country's capital and largest city...
(*Confidence change*: $0.099 \rightarrow 0.287 \rightarrow 0.352$)

**Mislabeled instance**: Lake **Managua** is a lake in Nicaragua. The Spanish name is Lago de Managua...
(*Confidence change*: $0.119 \rightarrow 0.037 \rightarrow 0.032$)

**Counterfactual instance**: **Managua** consists of more than one-sixth of the overall population of Nicaragua...

**Confusing instance**: The second largest lake in Nicaragua has an indigenous name Cocibolca (Sweet Sea)...

Table 4: A case of confidence updating in NQ dataset with counterfactual and confusing instances.

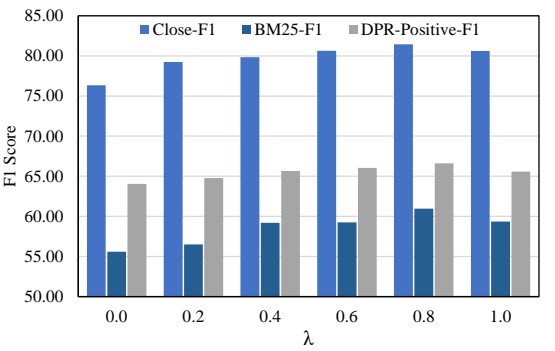

Figure 2: Model performances with different moving-average factor $\lambda$ on TriviaQA dataset with CDS-PA.

should be close to the negative samples in the representation space. Viewing them as positives will mislead the model learning. This shows the importance of estimating and leveraging label uncertainty in distant supervision, and our confidence-aware contrastive loss is effective for taking the uncertainty into consideration.

**Effects of hyper-parameter $\lambda$.** We analyse the sensitivity of moving-average factor $\lambda$ by using different values of $\lambda$. We test the model performance with $\lambda \in \{0.0, 0.2, 0.4, 0.6, 0.8, 1.0\}$ on TriviaQA dataset. From Figure 2, we can see that the model performances with Close, BM25 and DPR-Positive settings show similar trends on $\lambda$, and our method achieves the best results at $\lambda = 0.8$.

## 5.5 Case Study

To better demonstrate the effect of CDS, Table 4 shows an example of negative instance sampling and confidence estimation from the NQ dataset. The correct instance is the same as the manually annotated positive passage in the NQ dataset that describes the relation "Managua is the capital of Nicaragua". The mislabeled instance contains the answer "Managua", but instead of describing the city Managua, it is about Lake Managua which is not related to the question.

This case demonstrates the results of counterfactual instance sampling and confusing instance sampling. For the counterfactual instance, our method samples an instance that does not express the "*capital-of*" relation but contains the answer "Managua". For the confusing instance, our method retrieves a passage based on the mislabeled instance which does not contain the answer. This case shows that our counterfactual and confusing negative sampling algorithms can effectively sample high-quality and helpful negative instances for eliminating answer bias and denoising DS contexts.

For confidence estimation, in the beginning, the mislabeled instance has a higher confidence score than the correct instance. Along with the learning process, the confidence score of the mislabeled instance decreases to 0.032, and the correct instance gains the highest confidence for this <question, answer> pair. This demonstrates that CDS can successfully distinguish correct instances during training. By using these confidence scores as weights in the contrastive loss, our method can effectively leverage the uncertainty of DS-MRC instances.

## 6 Conclusion

This paper proposes Contrastive Distant Supervision – CDS, which can debias and denoise DS-MRC models by learning to distinguish confusing and noisy instances via confidence-aware contrastive learning. Specifically, CDS samples counterfactual negative instances to eliminate answer bias, and samples confusing negative instances to denoise noisy contexts. Then, a confidence-aware contrastive loss is proposed to estimate and

leverage the uncertainty of all DS instances during learning. Experimental results verify the effectiveness of CDS: On close-domain MRC, CDS significantly outperforms previous DS-MRC methods and achieves competitive performances with supervised MRC methods. On open-domain MRC settings, CDS even outperforms supervised MRC models by a large margin. Furthermore, contrastive distant supervision is model-free and thus can effectively complement other MRC techniques. We will release all codes, models and datasets on GitHub.

## Limitations

Current DS-MRC systems still need an external passage retrieval component, and we leave the fully end-to-end MRC systems for future work, i.e., the retrieval, the reading, and the answering are jointly modeled and learned. Besides, we use BM25 for passage retrieval and do not use recent dense retrieval models which may further increase the performance of our DS-MRC models.

The introduction of additional counterfactual and confusing instances will increase the time of training[2], because DS-MRC models need to learn to distinguish more contrastive and confusing instance pairs.

## Acknowledgements

We sincerely thank the reviewers for their insightful comments and valuable suggestions. This work is supported by the Natural Science Foundation of China (No.62122077 and 62106251). Hongyu Lin is sponsored by CCF-Baidu OpenFund.

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
