# OpenReview forum: "Contrastive Distant Supervision for Debiased and Denoised Machine Reading Comprehension"
_EMNLP/2023/Conference — EMNLP 2023 Findings_

### Official Review · Reviewer_dmb8 · 2023-08-01

**Soundness:** 3

**Excitement:**

3: Ambivalent: It has merits (e.g., it reports state-of-the-art results, the idea is nice), but there are key weaknesses (e.g., it describes incremental work), and it can significantly benefit from another round of revision. However, I won't object to accepting it if my co-reviewers champion it.

**Paper Topic And Main Contributions:**

This paper present a distant supervision based machine reading comprehension method by combing the contrastive learning and distant supervision method.

Specifically, in their method, they sample counterfactual negative instances and confusing negative instances, and use those sampled instances to do contrastive learning. Also, they modified the loss function, design some confidence estimation formulas.

Experimental results on naturalQA and TrivialQA show that their proposed method is effective and can even outperform supervised machine reading comprehension without any manual annotations.



**Questions For The Authors:**

Hi, thanks for this great submission, I have the following questions for you:

1: Can you provide more analysis examples showing the benefit of the proposed method? on what kind of examples, CDS is better, and on what examples, CDS is worse etc.

2: how sensitive is this method to hyper-parameters? in the paper you showed the performance w.r.t to different lambda, do you also have results for other parameters such as alpha, beta and K etc?

3: In this paper, you mention it outperforms the supervised methods, do you outperform the STOA supervised methods?

4: what's the future work in this direction?

**Reasons To Accept:**

Novel ideas and looks like strong experimental results.

**Reasons To Reject:**

Not sure whether this method is sensitive to hyper-parameter tuning or not.

**Reproducibility:**

3: Could reproduce the results with some difficulty. The settings of parameters are underspecified or subjectively determined; the training/evaluation data are not widely available.

**Reviewer Confidence:**

4: Quite sure. I tried to check the important points carefully. It's unlikely, though conceivable, that I missed something that should affect my ratings.

---

> ### Author Rebuttal · Authors · 2023-08-29
>
> Questions:
>
> > 1: Can you provide more analysis examples showing the benefit of the proposed method? on what kind of examples, CDS is better, and on what examples, CDS is worse etc.
>
> Answer:
> In Section 5.5 of our paper, we included a case study that highlights the benefits of the proposed method. This example demonstrates that the dynamic updating of confidence scores can benefit the model training process. We will provide additional analysis examples in the revised paper to better illustrate the benefits of our proposed method. We will include examples that showcase when CDS performs better and when it faces challenges compared to other approaches.
>
> ---
> > 2: how sensitive is this method to hyper-parameters? in the paper you showed the performance w.r.t to different lambda, do you also have results for other parameters such as alpha, beta and K etc?
>
> Answer:
> Due to space constraints of the paper, we were unable to include the results for other hyper-parameters such as alpha, beta, and K in the original paper. We recognize the importance of understanding the sensitivity of our method to these parameters. We have conducted thorough experiments exploring their effects and will ensure that these results are included in the final version of the paper.
>
> ---
> > 3: In this paper, you mention it outperforms the supervised methods, do you outperform the STOA supervised methods?
>
> Answer:
> Our primary focus is not a direct comparison with SOTA supervised methods. Comparing our method with supervised methods that use the same base pre-trained model is more fair. In our experiments, we used BERT as the base model for both the supervised and our proposed method. Furthermore, our proposed method achieves state-of-the-art results under distantly supervised reading comprehension tasks.
>
> ---
> > 4: what's the future work in this direction?
>
> Answer:
> In the era of large language models, a crucial direction is leveraging external documents and knowledge to enhance LLMs' background information during question answering. This process, similar to our approach, also faces challenges related to bias and noise. Our contrastive learning approach might inspire methods for fine-tuning and aligning LLMs to better incorporate external knowledge and provide more factual answers. This area requires continued exploration. In the revised paper, we will expand the "Future Work" section to discuss potential research directions that stem from our work.

---

### Official Review · Reviewer_GuVT · 2023-08-04

**Soundness:** 4

**Excitement:**

4: Strong: This paper deepens the understanding of some phenomenon or lowers the barriers to an existing research direction.

**Paper Topic And Main Contributions:**

This paper tackles two problems of Distantly Supervised Machine Reading Comprehension(DS-MRC): Answer Bias Problem and Context Noise Problem. The authors introduce Confidence-aware Contrastive Learning to address these issues, which is reported to be effective.

**Reasons To Accept:**

The addressed problems (bias and noise) are well-known issues that applies to various types of QA; the proposed method (Confidence-aware Contrastive Learning) seems to have a potential impact on these area. Also, the effectiveness of proposed method is solidly supported by experimental results.

**Reasons To Reject:**

The term 'counterfactual' in this paper is confusing; I am not sure if this is an appropriate usage of the concept. In this paper, 'counterfactual instance' means the 'uncertain' instances charactarized by lower model confidences.

However, as a logical concept, a conterfactual event is an event that is unactualized but possible to happen. I think there is a misalignment with this concept and the usage in this paper.

**Reproducibility:**

5: Could easily reproduce the results.

**Reviewer Confidence:**

3: Pretty sure, but there's a chance I missed something. Although I have a good feel for this area in general, I did not carefully check the paper's details, e.g., the math, experimental design, or novelty.

---

> ### Author Rebuttal · Authors · 2023-08-29
>
> > The term 'counterfactual' in this paper is confusing; I am not sure if this is an appropriate usage of the concept. In this paper, 'counterfactual instance' means the 'uncertain' instances charactarized by lower model confidences. However, as a logical concept, a conterfactual event is an event that is unactualized but possible to happen. I think there is a misalignment with this concept and the usage in this paper.
>
> Response:
>
> In our paper, we introduce the term “counterfactual instance” building on well-established concepts within the Visual Question Answering (VQA) domain. This term is particularly drawn from works like "Counterfactual samples synthesizing for robust visual question answering" , “Towards Causal VQA: Revealing and Reducing Spurious Correlations by Invariant and Covariant Semantic Editing”, and "Question conditioned counterfactual image generation for VQA", etc. Constructing counterfactual images as training instances is a common practice in VQA to reduce context bias and spurious correlations.
>
> Specifically, in our paper, a "factual" instance means the answer string from the passage is the accurate answer to the question, and a “counterfactual” instance refers to a scenario where, even if the answer is correct, it is presented in an irrelevant or incorrect context (the counterfactual context) given the question. Therefore, such answers should not be marked as the correct answer to the posed question. This term is consistent with the “counterfactual instance” in the VQA field. We will clarify this context in our revision to bridge any potential understanding gaps, ensuring that the meaning of “counterfactual instance” is clear to the readers.

---

### Official Review · Reviewer_6c51 · 2023-08-04

**Soundness:** 3

**Excitement:**

3: Ambivalent: It has merits (e.g., it reports state-of-the-art results, the idea is nice), but there are key weaknesses (e.g., it describes incremental work), and it can significantly benefit from another round of revision. However, I won't object to accepting it if my co-reviewers champion it.

**Paper Topic And Main Contributions:**

This paper introduces a new method of distant supervision in MRC based on the idea of contrastive learning, which includes both debiasing and denoising and needs no manual annotations. This method outperforms existing DS-MRC methods and even (under some settings) beats supervised methods.


**Questions For The Authors:**

1. Why do you choose a modified NQ dataset (Line 452-456) which is less frequently used to test models instead of the official version which is used by the majority?
2. The result of Supervised Model under DPR-Positive on TriviaQA in Table 1 seems strange compared to other results. Can you explain or analyze the reason of that extremely low result?

**Reasons To Accept:**

1. Clearly describes the mechanism of debiasing and denoising.
2. The description of the confidence-aware contrastive loss is clear and easy to understand.
3. An unsupervised method that can beat supervised methods is what everyone likes to see.

**Reasons To Reject:**

1. Lack of explanation and analysis of the model's deep mechanism. Why the dynamic update of confidence score works and why the model can outperformance supervised models so significantly need further detailed explanation. Only description but no explanation makes it short of interpretability.
2. Inconsistant symbol usage. The w_x in Eq. 5-7 and the w_i^t in Eq. 9 have different formats, which might lead to confusion.
3. An overview of the workflow and the model, which can make it easier to get the whole picture of the work, is needed.
4. Missing Ethics Statement (e.g., the reproductivity of the work).
5. Typo on Line 496 (wrong serial number '(2)').

**Reproducibility:**

4: Could mostly reproduce the results, but there may be some variation because of sample variance or minor variations in their interpretation of the protocol or method.

**Reviewer Confidence:**

4: Quite sure. I tried to check the important points carefully. It's unlikely, though conceivable, that I missed something that should affect my ratings.

---

> ### Author Rebuttal · Authors · 2023-08-29
>
> (1)
> > Lack of explanation and analysis of the model's deep mechanism. Why the dynamic update of confidence score works and why the model can outperformance supervised models so significantly need further detailed explanation. Only description but no explanation makes it short of interpretability.
>
> Response:
>
> Thank you for highlighting the importance of model interpretability. In Section 5.5, we provide a case study that shows an example to illustrate how the dynamic update of confidence scores influences the model's learning process. As shown in the example, along with the learning process, the confidence score of the mislabeled instance is decreased, and the correct instance gains the highest confidence. This demonstrates that CDS can successfully distinguish correct instances by dynamically updating the confidence scores.
>
> As to the significant outperformance of our models compared to supervised models when given retrieved passages, we believe this is because our models are trained within a distant supervision setting. Our models leverage retrieved passages which can learn more robust reading comprehension abilities, in contrast to supervised models which are trained solely on golden, high-quality passages. This further shows that our CDS method, after addressing the bias and noise issues of DS-MRC, can amplify the capabilities inherent in distant supervision.
>
> We acknowledge the need for more explicit explanations in these aspects and will ensure they are included in our revised paper.
>
> ---
> (2)
> > Inconsistant symbol usage. The w_x in Eq. 5-7 and the w_i^t in Eq. 9 have different formats, which might lead to confusion.
>
> Response:
>
> Thank you for pointing out the inconsistency in symbol usage. We apologize for any confusion this may cause. In our revised version, we will ensure consistent symbol formatting across equations to prevent any ambiguity.
>
> ---
> (3)
> > An overview of the workflow and the model, which can make it easier to get the whole picture of the work, is needed.
>
> Response:
>
> We acknowledge the importance of providing an overview of the workflow and the model and have illustrated the basic concept of our contrastive learning in Figure 1. Furthermore, we will incorporate an overview of the workflow and the model structure to enhance the clarity of our paper.
>
> ---
> (4)
> > Missing Ethics Statement (e.g., the reproductivity of the work).
>
> Response:
>
> We will include a comprehensive Ethics Statement in our paper, addressing concerns related to reproducibility and ethical considerations.
>
> ---
> (5)
> > Typo on Line 496 (wrong serial number '(2)').
>
> Response:
>
> Thank you for identifying the typo. We will carefully review and correct this mistake in the revision.
>
> ---
> Questions:
> > 1. Why do you choose a modified NQ dataset (Line 452-456) which is less frequently used to test models instead of the official version which is used by the majority?
>
> Answer:
> We selected the modified NQ dataset provided by Sen and Saffari(2020) due to its alignment with the extractive MRC setup of our study. This dataset is a subset of NQ and the MRC task is defined as finding the short answer in the long answer. The modified NQ dataset offers a more standard context for MRC tasks, whereas the official NQ dataset involves reading long documents or multi-document retrieval, which does not align with our focus. Therefore, the modified NQ dataset provides a suitable setting for evaluating our model's performance. Using this modified NQ dataset is a common practice for extractive MRC experiments.
>
> > 2. The result of Supervised Model under DPR-Positive on TriviaQA in Table 1 seems strange compared to other results. Can you explain or analyze the reason of that extremely low result?
>
> Answer:
> The unusual result of the Supervised Model under DPR-Positive on TriviaQA in Table 1 can be attributed to differences in document sources between our study and the TriviaQA dataset. TriviaQA's document collection involves web search results and Wikipedia articles (please refer to the original TriviaQA paper for details), while we solely relied on retrieving Wikipedia articles for the MRC passages. This difference in passage sources can explain the discrepancy in results.

---

### Meta-Review · Area_Chair_za8q · 2023-09-15

**Recommendation:** 3

**Metareview:**

This paper presents a distant supervision method for training MRC models using a contrastive loss, with the primary contribution being a model confidence-aware sampling strategy for obtaining negative samples to make training more effective. The reviewers all agreed that the proposed method is sound, that the results are strong, and that the paper is well-written. They recommended that the authors include further analysis of the trained model's errors and sensitivity to hyperparameter choices.

---

### Decision · Program_Chairs · 2023-10-07

**Decision:**

Accept-Findings

**Comment:**

This paper presents a distant supervision method for training MRC models using a contrastive loss, with the primary contribution being a model confidence-aware sampling strategy for obtaining negative samples to make training more effective. The reviewers all agreed that the proposed method is sound, that the results are strong, and that the paper is well-written. They recommended that the authors include further analysis of the trained model's errors and sensitivity to hyperparameter choices.